# Immune-Based Combinations versus Sorafenib as First-Line Treatment for Advanced Hepatocellular Carcinoma: A Meta-Analysis

**Alessandro Rizzo** [1,*,†] ![ID], **Angela Dalia Ricci** [2,†] ![ID], **Annarita Fanizzi** [3] ![ID], **Raffaella Massafra** [3], **Raffaele De Luca** [4] and **Giovanni Brandi** [5,6]

1   Struttura Semplice Dipartimentale di Oncologia Medica per la Presa in Carico Globale del Paziente Oncologico "Don Tonino Bello", I.R.C.C.S. Istituto Tumori "Giovanni Paolo II", Viale Orazio Flacco 65, 70124 Bari, Italy

2   Medical Oncology Unit, National Institute of Gastroenterology, "Saverio de Bellis" Research Hospital, 70013 Castellana Grotte, Italy

3   Struttura Semplice Dipartimentale di Fisica Sanitaria, I.R.C.C.S. Istituto Tumori "Giovanni Paolo II", Viale Orazio Flacco 65, 70124 Bari, Italy

4   Department of Surgical Oncology, IRCCS Istituto Tumori "Giovanni Paolo II", 70124 Bari, Italy

5   Department of Specialized, Experimental and Diagnostic Medicine, University of Bologna, Via Giuseppe Massarenti, 9, 40138 Bologna, Italy

6   Division of Medical Oncology, IRCCS Azienda Ospedaliero-Universitaria di Bologna, Via Albertoni, 15, 40138 Bologna, Italy

*   Correspondence: rizzo.alessandro179@gmail.com

†   These authors contributed equally to this work.

**Abstract:** Recent years have observed the emergence of novel therapeutic opportunities for advanced hepatocellular carcinoma (HCC), such as combination therapies including immune checkpoint inhibitors. We performed a meta-analysis with the aim to compare median overall survival (OS), median progression-free survival (PFS), complete response (CR) rate, and partial response (PR) rate in advanced HCC patients receiving immune-based combinations versus sorafenib. A total of 2176 HCC patients were available for the meta-analysis (immune-based combinations = 1334; sorafenib = 842) and four trials were included. Immune-based combinations decreased the risk of death by 27% (HR, 0.73; 95% CI, 0.65–0.83; $p < 0.001$); similarly, a PFS benefit was observed (HR, 0.64; 95% CI, 0.5–0.84; $p < 0.001$). In addition, immune-based combinations showed better CR rate and PR rate, with ORs of 12.4 (95% CI, 3.02–50.85; $p < 0.001$) and 3.48 (95% CI, 2.52–4.8; $p < 0.03$), respectively. The current study further confirms that first-line immune-based combinations have a place in the management of HCC. The CR rate observed in HCC patients receiving immune-based combinations appears more than twelve times higher compared with sorafenib monotherapy, supporting the long-term benefit of these combinatorial strategies, with even the possibility to cure advanced disease.

**Keywords:** hepatocellular carcinoma; atezolizumab; bevacizumab; immunotherapy; cabozantinib; sorafenib

## 1. Introduction

Hepatocellular carcinoma (HCC) represents the most diagnosed type of primary liver cancer and accounts for more than 80% of all cases of primary liver tumors [1]. The results achieved by the use of immune checkpoint inhibitors (ICIs) in several tumor types have led to a similar development in HCC [2]. Two anti-PD-1 antibodies used as monotherapy, pembrolizumab, and nivolumab, were approved by the United States FDA [3,4]. At the same time, the two confirmatory phase III trials comparing nivolumab versus sorafenib as front-line treatment and pembrolizumab versus placebo in previously treated patients, respectively—did not meet their primary endpoints [5,6].

Subsequent results of clinical trials assessing combinatorial strategies including ICIs plus other anticancer agents, and especially antiangiogenic drugs, have been more striking and have marked a new era in HCC management [7]. The IMbrave150 trial compared the combination of the anti-PD-L1 atezolizumab plus the antiangiogenic agent bevacizumab versus sorafenib monotherapy in previously untreated advanced HCC patients [8,9]; of note, the results of this phase III study have led to the approval of atezolizumab—bevacizumab due to a median overall survival (OS) of 19.2 months compared with 13.4 months for sorafenib monotherapy (Hazard Ratio [HR], 0.58; 95% Confidence Interval [CI], 0.42–0.79). Similarly, the study reported a median progression free survival (PFS) benefit and higher overall response rate (ORR) in patients treated with the immune-based combination. The combination of atezolizumab plus bevacizumab is currently considered the new standard of care in first-line HCC and has been approved in several countries worldwide. Other immune-based combinations have been tested and are currently being explored. Among these, the COSMIC-312 trial compared the combination of atezolizumab plus cabozantinib versus sorafenib as a first-line treatment for patients with advanced disease [10], and no difference in OS was highlighted. In another phase II/III trial of ORIENT-32, the investigators compared the combination of the PD-1 inhibitor sintilimab plus a bevacizumab biosimilar (IBI305) versus sorafenib alone, reporting a statistically significant PFS and OS improvement in patients treated with sintilimab—IBI305 [11].

The role of another immune-based combination including two ICIs, durvalumab, and the anti-CTLA-4 antibody tremelimumab, has been explored in the HIMALAYA trial [12]. In this phase III trial, the median OS was 16.4 months in patients receiving durvalumab plus tremelimumab versus 13.8 months in the sorafenib monotherapy arm. Based on these premises, we performed a meta-analysis aimed at comparing OS, PFS, complete response (CR) rate, and partial response (PR) rate in HCC patients receiving immune-based combinations versus sorafenib monotherapy as first-line treatment.

## 2. Materials and Methods

### 2.1. Search Strategy

All phase II and III clinical trials published from 15 June 2008 to 29 July 2022, comparing immune-based combinations versus sorafenib monotherapy in treatment-naïve advanced/metastatic HCC were retrieved by three different authors. Keywords used for searching on PubMed/Medline, Cochrane Library, and EMBASE were the following: "atezolizumab" OR "nivolumab" OR "pembrolizumab" OR "ipilimumab" OR "avelumab" OR "durvalumab" OR "tremelimumab" OR "sintilimab" OR "immune checkpoint inhibitors" AND "sorafenib" AND "first-line treatment" AND "hepatocellular carcinoma" OR "HCC". Only articles published as full-text in peer-reviewed journals and written in the English language were included. In addition, proceedings of the main international oncological meetings (European Society of Medical Oncology [ESMO], American Society of Clinical Oncology [ASCO], American Association for Cancer Research [AACR], European CanCer Organization [ECCO]) were also searched from 2008 onward for relevant abstracts.

### 2.2. Selection Criteria

Studies selected from the first analysis were subsequently restricted to: (1) prospective phase II and III randomized controlled trials (RCTs) in advanced/metastatic HCC; (2) participants enrolled in first-line treatment with immune-based combinations versus sorafenib; (3) studies with available data in terms of OS, PFS, CR rate, and PR rate.

### 2.3. Data Extraction

The following data were extracted for each publication: (1) study general information (author, year, phase, carry out the country); (2) interventions and dosage; (3) number of patients; (4) primary outcomes; (5) available outcomes in terms of OS, PFS, CR rate, PR rate, and DCR rate. Three separate authors conducted the search and identification independently. RECIST 1.1. criteria were used. The meta-analysis was conducted according to

Preferred Reporting Items for Systematic Review and Meta-Analyses (PRISMA) guidelines (File S1) [13].

### 2.4. Risk of Bias Assessment in Included Studies

The methodological quality of the included trials was evaluated using the Cochrane Collaboration tool and the risk of bias in the selected studies was assessed independently by three authors [14]. Studies examined were graded as having a "low risk", "high risk", or "unclear risk" of bias across the specified domains of selection, performance, attrition, and reporting bias. The lists of outcomes reported in the published papers were compared to those from study protocols or trial registries. The results of the assessment were summarized in a risk of bias graph (Figure 1).

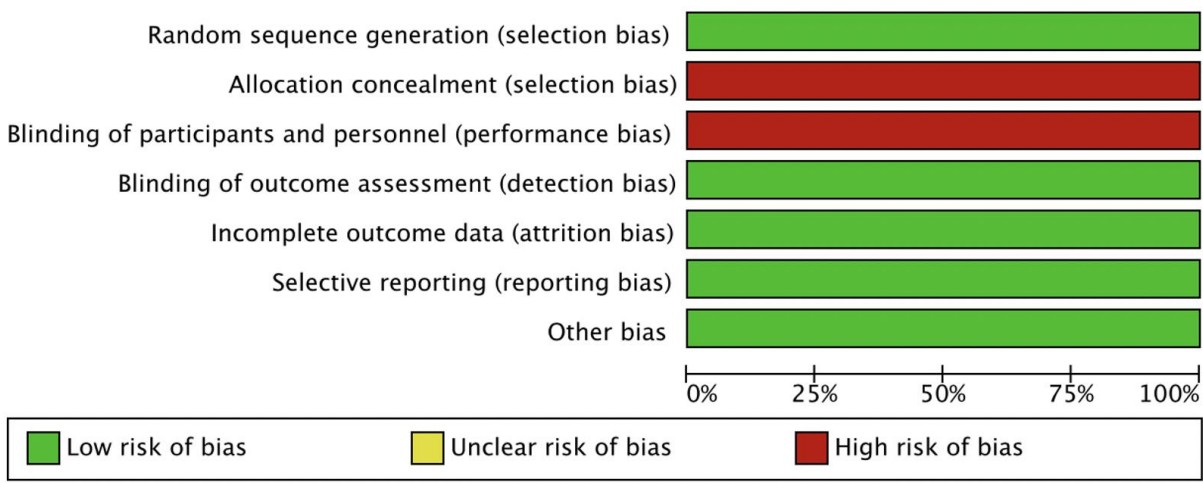

**Figure 1.** Risk of bias graph; Authors' judgments about each risk of bias item are presented as percentages across all included studies.

### 2.5. Types of Outcomes Measures

We examined OS, PFS, CR rate, and PR rates. For each trial, three different authors extracted data from the safety analysis. Data were obtained from each study.

### 2.6. Statistical Design

All statistical analyses were performed using ProMeta 3 software and R Studio.

Effect measures for OS and PFS were HRs and 95% CIs, which were extracted from available studies. Similarly, ORs were used to analyze dichotomous variables, including CR rate, PR rate, and DCR rate. Forest plots were used to assess HRs to describe the relationship between treatment and OS and PFS in the specified cohorts of patients; forest plots were also used to assess ORs for CR rate, and PR rate.

Statistical heterogeneity between trials was examined using the Chi-square test and the $I^2$ statistic; substantial heterogeneity was considered to exist when the $I^2$ value was greater than 50% or there was a low *p*-value (<0.10) in the Chi-square test [15]. When no heterogeneity was noted, the fixed effects model was used, while the random effects model was applied in the presence of significant heterogeneity. Presence of publication bias was formally evaluated using funnel plots.

### 3. Results

#### 3.1. Selected Studies

Our search resulted in the identification of 2117 potentially relevant reports, which were subsequently restricted to four following independent evaluations by three authors [9–12]. We excluded 2113 records as non-pertinent reports (systematic reviews and meta-analyses, editorial, case reports, review articles, pre-clinical studies, retrospective studies, single-arm

trials, non-randomized trials, ongoing studies/trials in progress). All studies included in our analysis (Table 1) were judged as having a low risk of bias in separate reviews of three authors. Figure 2 shows the search process. The above RCTs compared immune-based combinations versus sorafenib as first-line treatment in HCC patients with advanced disease. A summary of the included trials is presented in Table 1 [9–12]. A total of 2176 HCC patients were available for the meta-analysis (ICIs = 1334; sorafenib = 842).

**Table 1.** Summary of all the included studies in the present meta-analysis. Abbreviations: pts: patients; RCT: randomized controlled clinical trial.

| Trial | Study Design | Demographics (Arm A; Arm B) | No. pts Arm A/Arm B |
|---|---|---|---|
| IMbrave150 [9] | Phase 3 RCT | Median Age: 64; 66<br>Male sex: 82%; 83%<br>Asian patients: 40%, 41%<br>ECOG-PS 0: 62%; 62%<br>HBV-positive: 49%; 46%<br>HCV-positive: 21%; 22% | Atezolizumab plus bevacizumab: 326<br><br>Sorafenib: 159 |
| COSMIC-312 [10] | Phase 3 RCT | Median Age: 65; 64<br>Male sex: 86%; 88%<br>Asian patients: 25%, 27%<br>ECOG-PS 0: 65%; 61%<br>HBV-positive: 30%; 29%<br>HCV-positive: 28%; 28% | Atezolizumab plus cabozantinib: 250<br><br>Sorafenib: 122 |
| ORIENT-32 [11] | Phase 2/3 RCT | Median Age: 53; 654<br>Male sex: 82%; 83%<br>Asian patients: 40%, 41%<br>ECOG-PS 0: 62%; 62%<br>HBV-positive: 49%; 46%<br>HCV-positive: 21%; 22% | Sintilimab plus IBI305: 365<br><br>Sorafenib: 172 |
| HIMALAYA [12] | Phase 3 RCT | Median Age: 65; 64<br>Male sex: 83.2%; 86.6%<br>Asian patients: 39.7%, 40.1%<br>ECOG-PS 0: 62.1%; 62.0%<br>HBV-positive: 31.0%; 30.6%<br>HCV-positive: 28.0%; 26.7% | Durvalumab plus tremelimumab: 393<br><br>Sorafenib: 389 |

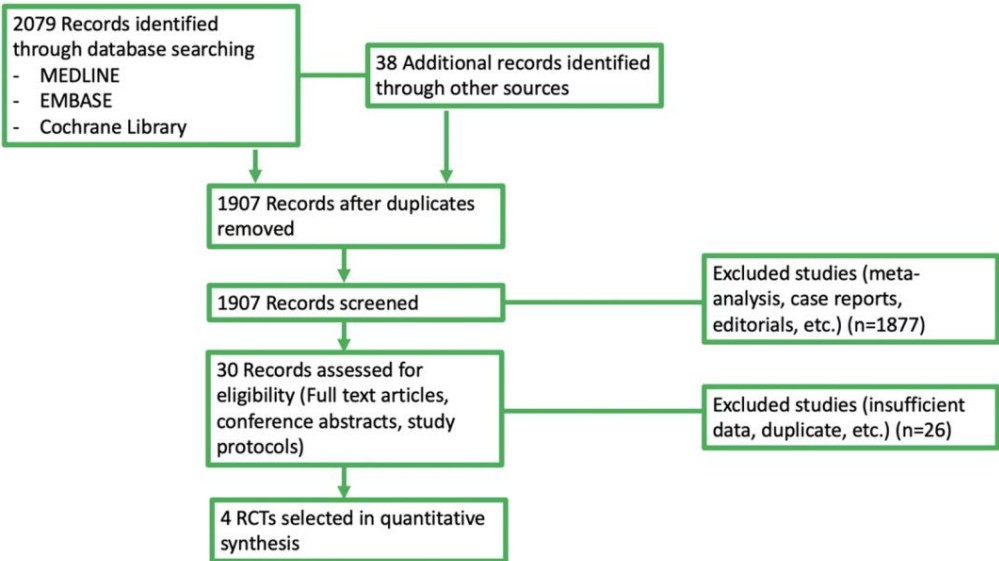

**Figure 2.** Diagram of all the trials included and excluded in the present meta-analysis.

### 3.2. Overall Survival

The pooled HR for OS was 0.73 (95% CI, 0.65–0.83; $p < 0.001$) (Figure 3), corresponding to a risk of death decrease by 27% in patients receiving immune-based combinations compared to sorafenib monotherapy; the analysis was associated to low heterogeneity ($I^2$ of 21%), and thus, a fixed-effects model was used.

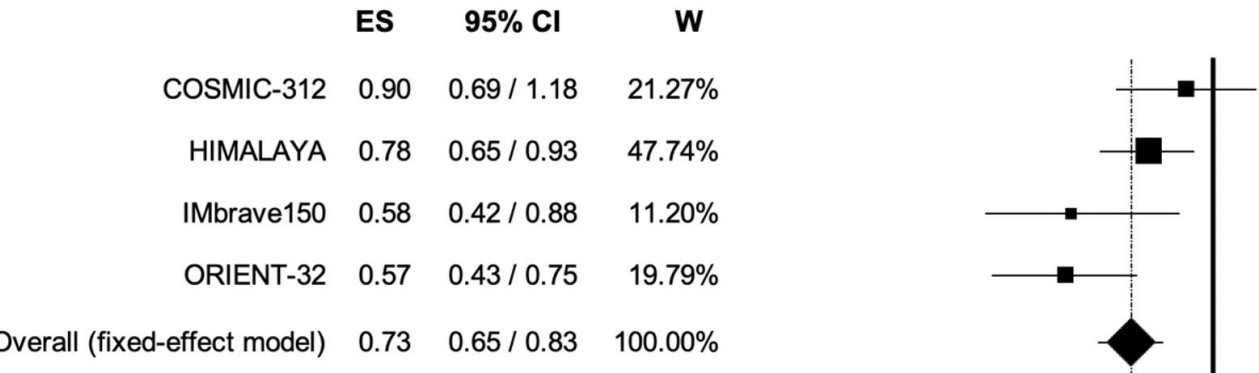

**Figure 3.** Forest plot of comparison between immune-based combinations versus sorafenib monotherapy; the outcome was Hazard Ratio (HR) of median Overall Survival. CI: confidence interval; ES: effect size; W: weight.

### 3.3. Progression-Free Survival

The pooled HR for PFS was 0.64 (95% CI, 0.50–0.84; $p < 0.001$) (Figure 4) in patients treated with immune-based combinations compared to sorafenib single-agent; the analysis was associated with high heterogeneity ($I^2$ of 85%). The analysis was conducted using a random-effects model.

|  | ES | 95% CI | W |
|---|---|---|---|
| COSMIC-312 | 0.56 | 0.46 / 0.70 | 24.81% |
| HIMALAYA | 0.90 | 0.77 / 1.05 | 26.74% |
| IMbrave150 | 0.59 | 0.47 / 0.76 | 23.64% |
| ORIENT-32 | 0.56 | 0.46 / 0.70 | 24.81% |
| Overall (random-effects model) | 0.64 | 0.50 / 0.84 | 100.00% |

**Figure 4.** Forest plot of comparison between immune-based combinations versus sorafenib monotherapy; the outcome was Hazard Ratio (HR) of median Progression-Free Survival. CI: confidence interval; ES: effect size; W: weight.

### 3.4. Complete Response Rate

Immune-based combinations showed a better CR rate, with OR of 12.4 (95% CI, 3.02–50.85; $p < 0.001$) (Figure 5), compared to sorafenib monotherapy. The analysis reported low heterogeneity ($I^2$ of 0%), and thus, a fixed-effects model was used.

### 3.5. Partial Response Rate

A higher PR rate was observed in patients treated with immune-based combinations compared to sorafenib, with pooled OR for a PR rate of 3.48 (95% CI, 2.52–4.8; $p < 0.03$) (Figure 6). Low heterogeneity was detected in the analysis ($I^2$ of 26%), and a fixed-effects model was used.

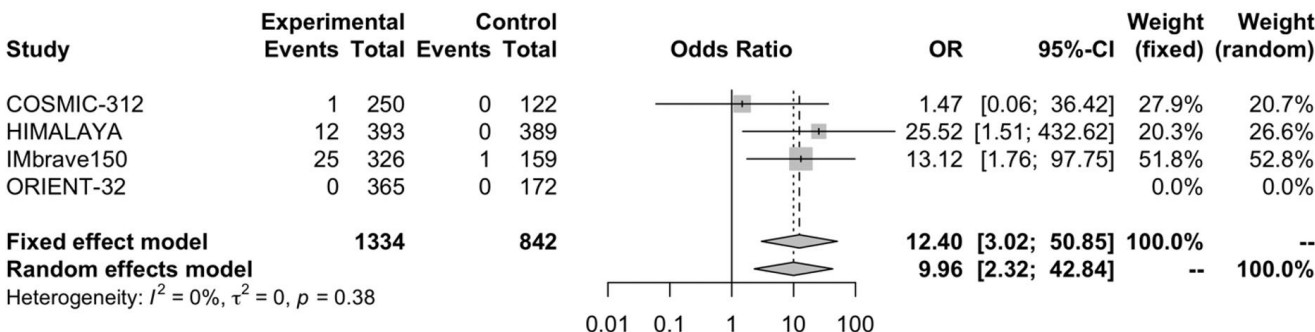

**Figure 5.** Forest plot of comparison between immune-based combinations versus sorafenib monotherapy; the outcome was the Odds Ratio (OR) of complete response rate. CI: confidence interval; OR: Odds Ratio.

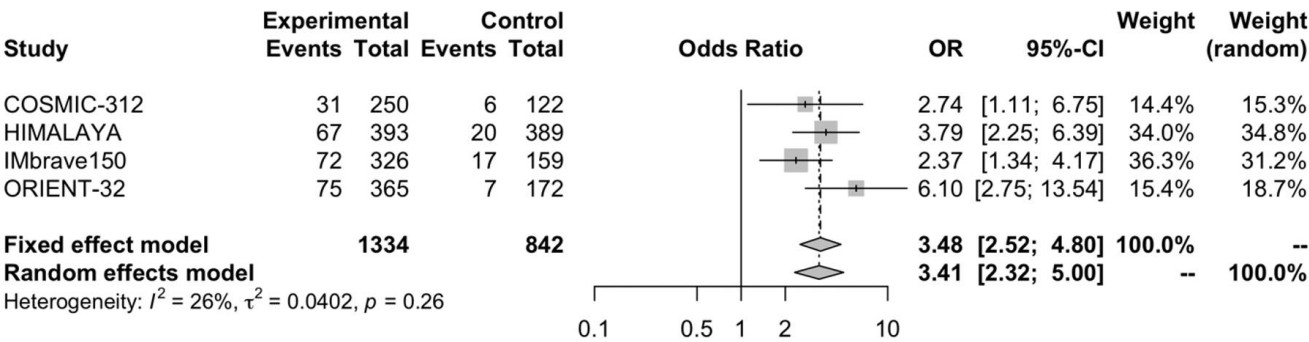

**Figure 6.** Forest plot of comparison between immune-based combinations versus sorafenib monotherapy; the outcome was the Odds Ratio (OR) of partial response rate. CI: confidence interval; OR: Odds Ratio.

## 4. Discussion

Herein, we performed a meta-analysis aimed at comparing clinical outcomes in advanced HCC patients receiving first-line immune-based combinations versus sorafenib. In our analysis, we included four RCTs—IMbrave150, HIMALAYA, COSMIC-312, and ORIENT-32—as a whole encompassing 2176 HCCs. According to the results of the meta-analysis, patients treated with immune-based combinations reported longer median OS and PFS, as well as a higher CR rate and PR rate. Of note, the CR rate observed in HCC patients receiving immune-based combinations appears more than twelve times higher compared with sorafenib monotherapy, supporting the long-term benefit of these combinatorial strategies, with even the possibility to cure advanced HCC. The present meta-analysis holds its own strengths and limitations. The strengths of our analysis encompass the inclusion of only phase II and III RCTs [9–12] and the overall number of patients (2176; immune-based combinations arm = 1334, sorafenib monotherapy arm = 842). Nonetheless, the results of the meta-analysis should be interpreted with caution, due to the presence of some limitations. First, individual patient data were not available, and thus, aggregate data included in our analysis were extracted from clinical trials results. In addition, our analysis did not include safety results or subgroup data, since our aim was mainly to focus on efficacy outcomes, especially in terms of the CR rate and PR rate. Second, the included clinical trials explored four different immune-based combinations, such as atezolizumab plus bevacizumab, atezolizumab plus cabozantinib, sintilimab plus IBI305, and the double checkpoint blockade with durvalumab plus tremelimumab [9–12]. Of note, these therapeutic strategies present distinct and not superimposable efficacy and safety profiles, something that should be kept in mind when interpreting the results of this aggregate analysis. In addition, differences in demographics across studies should be considered. An example is represented by the lower proportion of Asian patients in COSMIC-312 compared to HIMALAYA, IMbrave150, and

ORIENT-32, as reported in Table 1 [9–12]. Conversely, a higher proportion of HBV-positive patients was included in the Imbrave150 (49% of the experimental arm and 46% of the control arm) and ORIENT-32 (49% and 46% in the immune-based combination and the sorafenib group, respectively), compared to COSMIC-312 and HIMALAYA [9–12]. Another key point to consider is the heterogeneity in terms of available clinical outcomes; for example, COSMIC-312 reported the final PFS analysis but only the interim OS analysis, and this study was the only trial highlighting no significant OS benefit. In addition, despite the immune-based combinations being associated with higher CR rates, the distribution is not homogeneous. In fact, as reported in Figure 5, only one case of CR was observed in COSMIC-312 while no CRs in ORIENT-32. Similarly, no PFS improvement was observed in HIMALAYA. These further elements should be kept in mind when discussing the results of the current meta-analysis.

Impressive steps forward have been recently made in the field of systemic therapies for advanced HCC [16]. Atezolizumab plus bevacizumab represents the first treatment that has been proven to be more effective than sorafenib monotherapy in the practice—changing IMbrave150 phase III trial and has opened the doors of a new era in HCC. In addition, other immune-based combinations have been tested and are currently under assessment, with these combinatorial strategies showing prolonged OS, PFS, and higher CR rates and PR rates, as also confirmed by our meta-analysis [17]. In this scenario, immunotherapy seems to have finally found its role for advanced HCC, with the association of ICIs plus antiangiogenic agents or with ICIs targeting different pathways representing exciting strategies [18,19]. Without any direct comparison of first-line immune-based combinations, clinicians are called to choose according to differences in inclusion criteria and demographics between clinical trials, as well as considering the safety profiles of these combinations.

Several questions remain unanswered. Among these, the choice of the optimal treatment sequencing is a matter of debate, given the paucity of studies investigating second-line treatment post-ICIs. Moreover, in a setting that remains palliative, the safety profile is a key point to consider, which is of notable importance from the patient perspective. Despite all trials reporting manageable toxicities for HCC patients receiving immune-based combinations, some points should be highlighted. Since HCC patients frequently have concurrent diseases (e.g., liver cirrhosis, metabolic alterations) and poor liver function, careful attention should be given to patients' comorbidities, taking into account the possible contraindications for ICIs in cancer patients. Moreover, the lack of validated biomarkers of response represents an important issue since only a proportion of HCC patients benefit from immunotherapy [20]. Based on these premises, a greater understanding of the role of potential biomarkers including PD-L1 expression, tumor mutational burden (TMB), microsatellite instability (MSI) status, gut microbiota, and several others is fundamental. For example, HCC etiology has been also suggested to influence response to ICIs, with preclinical studies showing an intrinsic resistance to anti-PD-1 treatment in mice models of NASH-related HCC, with this evidence which has been supported by a landmark study showing a reduced benefit of ICIs in non-viral related HCC [21,22]. Some driver mutations have been suggested with the etiology of HCC, as in the case of HBV-related TP53 mutations and CTNNB1 in alcohol abuse [23–25]. In addition, information regarding the tumor microenvironment (TME) may provide further insights into HCC immunotherapy. Treatment regimens, including immune-based combinations, may reshape the HCC TME by inducing an adaptive response in cell components that reflects altered transcriptomes and proteomes. The adaptive response not only increases HCC survival, progression, and metastasis under therapeutic pressure but may also provide a key for novel treatments, including immunotherapy and antiangiogenic agents. Further efforts are needed to implement precision HCC immunotherapy, where HCC etiology may play a relevant role. Stratification of patients based on HCC etiology in future clinical trials is warranted, and the next few years will tell us how these novel agents and immune-based combinations may impact the treatment scenario of this deadly malignancy with many unanswered questions.

## 5. Conclusions

The current meta-analysis confirms that front-line immune-based combinations have an important place in the management of HCC. The CR rate observed in HCC patients receiving immune-based combinations appears more than twelve times higher compared with sorafenib monotherapy, supporting the long-term benefit of these combinatorial strategies, with even the possibility to cure advanced disease.

**Supplementary Materials:** The following supporting information can be downloaded at: https://www.mdpi.com/article/10.3390/curroncol30010057/s1, File S1: Preferred Reporting Items for Systematic Review and Meta-Analyses (PRISMA) guidelines.

**Author Contributions:** Conceptualization, all authors; methodology, all authors; software, all authors; validation, all authors; formal analysis, all authors; investigation, all authors; resources, all authors; data curation, A.R.; writing—original draft preparation, A.R.; writing—review and editing, all authors; visualization, all authors; supervision, all authors; project administration, all authors; funding acquisition, all authors. All authors have read and agreed to the published version of the manuscript.

**Funding:** This research received no external funding.

**Institutional Review Board Statement:** Not applicable.

**Informed Consent Statement:** Not applicable.

**Data Availability Statement:** Not applicable.

**Conflicts of Interest:** The authors declare no conflict of interest.

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
