# Peer review of "Immune-Based Combinations versus Sorafenib as First-Line Treatment for Advanced Hepatocellular Carcinoma: A Meta-Analysis"

_curroncol, doi:10.3390/curroncol30010057_

Round 1
Reviewer 1 Report
The authors present the systematic review and metaanalysis investigating the OS, PFS and response rate in HCC patients receiving immune-based combinations versus sorafenib monotherapy as first-line treatment. The current study further confirms that first-line immune-based combinations should be part initial treatment advanced HCC patients. The authors have performed an extensive literature search and summarized the most recent and relevant papers on the subject. The text is well written and easy to read and follow it. I would like to offer the following points for consideration by the authors towards the improvement of the manuscript:
1- Since there is a newly presented study, it would be good to update the search range and repeat the analysis due to limited number of studies for meta-analysis.
- LBA35 - Camrelizumab (C) plus rivoceranib (R) vs. sorafenib (S) as first-line therapy for unresectable hepatocellular carcinoma (uHCC): A randomized, phase III trial
2- Was the meta analysis submitted on PROSPERO or OSF? If not, why? If yes, please provide the reference number.
3- Please check the accuracy of HR for PFS in ORIENT-32 study (HR: 1.00 95%CI: 0.90-1.10)
4- P6 Line 199 and P7 Line 223 : “ Sunitinib” ?
5- It is not clear exactly how the authors narrowed the number of articles from over 1900 to 30. Please elaborate on the exclusion criteria with numbers .
6- According to the previous studies by Schulze et al. (Nat Genet 2015), the driver mutations are closely associated with the etiology of HCC; for example, TP53 related to HBV infection, and CTNNB1 related to alcohol consumption. Since there are several molecular and immunological subtype classifications for the implementation of precision immune oncology in HCC (Sia et al. Gastroenterology 2017, Shimada et al. EBioMedicine 2019, Int J Clin Oncol 2022, Llovet et al. Nat Rev Clin Oncol 2022), the authors must discuss the role of molecular subtyping oncology in HCC etiology and immunotherapy. Subgroup analyzes may be performed according to the etiology of HCC.
7- Please include more information on the effects of combined immunotherapy in terms of the tumor immune microenvironment. Namely, the authors should clarify the additional benefit of combined immunotherapy.
Author Response
Dear Reviewer,
Thank you for the time spent revising our paper. We are really grateful for your comments.
- Thank you for this suggestion. However, in the current analysis we included only studies published as full-text in peer-reviewed journals, and thus, the study was excluded.
- No, the current meta-analysis was not registered on PROSPERO.
- Thank you for this important suggestion and thank you for catching this oversight. We have re-performed the analysis since there was a mistake (orange).
- We modified accordingly.
- Thank you for this comment. The exclusion of duplicates and other studies was specified in Figure 2, as required.
- Thank you for these important suggestions. We largely modified the Discussion section, by discussing the topics above and by including the referenced papers.
Thank you again. We hope the revised paper will better suit the journal.
Reviewer 2 Report
Considering that all four trials included in this meta-analysis were positive trials comparing immune-based combinations with sorafenib, the results were in line with each trial and showed the superiority of immune-based combinations. Still, I believe this is a needed addition to the literature. However, I think there are some points to improve:
-Please check for typos. For example, on page 4, line 152 "advances".
-Objective response and complete response are compared, but it should be clearly given if RECIST 1.1 was used.
- On page 7, line 223: "sunitinib"?
-COSMIC-312 reported the final PFS analysis but only the interim OS analysis. Please discuss this situation, considering it is the only study with no significant OS benefit.
- Despite the fact that immune-based combinations have a higher complete response rate, the distribution is not homogenous. For example, in ORIENT-32 no CR, and in COSMIC-312 only one. Similarly, despite the OS benefit, there is no improvement in PFS in the HIMALAYA trial. This situation should be elaborately discussed.
- The last paragraph of the discussion is too general and not mainly related to current results.
- Limitations: No safety analysis, no subgroup analysis for PFS or objective response.
Author Response
Dear Reviewer,
Thank you for the time spent revising our paper and thank you for your comments. We modified accordingly, throughout the manuscript.
- Thank you for catching this oversight. We corrected this typo and other similar mistakes in the paper, as suggested (orange).
- We better specified this point (orange).
- Sorry for this oversight. We modified accordingly.
- We discuss this issue and we modified the Discussion section, as required.
- Thank you. We further reported the limitations of the current paper.
Thank you again for your suggestions. We hope the revised manuscript will better suit the journal.
Round 2
Reviewer 1 Report
I am satisfied that the authors have addressed all of my previous concerns about the article. It is now much improved and I feel that it is now suitable for publication.